# Evaluating the impact of the nationwide public–private mix (PPM) program for tuberculosis under National Health Insurance in South Korea: A difference in differences analysis

Sarah Yu[1,2,3‡], Hojoon Sohn[4‡], Hae-Young Kim[2,3], Hyunwoo Kim[1,5], Kyung-Hyun Oh[1,6], Hee-Jin Kim[1], Haejoo Chung[2,3], Hongjo Choi[1,7]*

1 Korean Institute of Tuberculosis, Korean National Tuberculosis Association, Cheongju, Republic of Korea, 2 School of Health Policy & Management, College of Health Science, Korea University, Seoul, Republic of Korea, 3 BK21 FOUR R&E Center for Learning Health Systems, Korea University, Seoul, Republic of Korea, 4 Department of Epidemiology, Johns Hopkins Bloomberg School of Public Health, Baltimore, Maryland, United States of America, 5 Department of Health Research Methods, Evidence, and Impact, McMaster University, Hamilton, Ontario, Canada, 6 End TB and Leprosy Unit, World Health Organization Regional Office for the Western Pacific, Manila, Philippines, 7 Department of Preventive Medicine, College of Medicine, Konyang University, Daejeon, Republic of Korea

‡ Contributed equally to this manuscript.
* hongjo@konyang.ac.kr

**Data Availability Statement:** Data cannot be shared publicly because of the data sharing policies at the Korea Centers for Disease Control and

## Abstract

### Background

Public–private mix (PPM) programs on tuberculosis (TB) have a critical role in engaging and integrating the private sector into the national TB control efforts in order to meet the End TB Strategy targets. South Korea's PPM program can provide important insights on the long-term impact and policy gaps in the development and expansion of PPM as a nationwide program.

### Methods and findings

Healthcare is privatized in South Korea, and a majority (80.3% in 2009) of TB patients sought care in the private sector. Since 2009, South Korea has rapidly expanded its PPM program coverage under the National Health Insurance (NHI) scheme as a formal national program with dedicated PPM nurses managing TB patients in both the private and public sectors. Using the difference in differences (DID) analytic framework, we compared relative changes in TB treatment outcomes—treatment success (TS) and loss to follow-up (LTFU)—in the private and public sector between the 2009 and 2014 TB patient cohorts. Propensity score matching (PSM) using the kernel method was done to adjust for imbalances in the covariates between the 2 population cohorts. The 2009 cohort included 6,195 (63.0% male, 37.0% female; mean age: 42.1) and 27,396 (56.1% male, 43.9% female; mean age: 45.7) TB patients in the public and private sectors, respectively. The 2014 cohort included 2,803

Prevention Agency (KCDA), governed by the Statistics Act Enforcement Ordinance Article 17 Item 2 ("Procedures and Methods for Statistics-based Policy Evaluation" available at: http://kostat.go.kr/portal/eng/aboutUs/5/6/index.static). Data are available only for researchers who meet the criteria for access to confidential data, subject to approval from respective ethics review committee at the KCDA. Queries for data access can be made to Ms. Kim, Jinsun at KCDA (http://kdca.go.kr/index.es?sid=a3).

**Funding:** This work was supported by the Korea Disease Control and Prevention Agency (KDCA), funded by the Ministry of Health & Welfare, Republic of Korea (grant number: 2017-E3100-200) (HC) and a grant of the Korea Health Technology R&D Project through the Korea Health Industry Development Institute (KHIDI), funded by the Ministry of Health & Welfare, Republic of Korea (grand number: HI19C1235) (HC). KDCA was provided the notification data from the Korean National Tuberculosis Surveillance System (KNTSS). The funders had no role in study design, data collection and analysis, decision to publish, or preparation of the manuscript.

**Competing interests:** The authors have declared that no competing interests exist.

**Abbreviations:** CI, confidence interval; CXR, chest X-ray; DID, difference in differences; DR, drug-resistant; EP, extrapulmonary; ICD-10, International Classification of Diseases-10th Revision; KCDC, Korean Centers for Disease Control and Prevention; KNTSS, Korean National Tuberculosis Surveillance System; LTFU, loss to follow-up; NHI, National Health Insurance; NTP, National Tuberculosis Program; PPM, public–private mix; PSM, propensity score matching; STROBE, Strengthening the Reporting of Observational Studies in Epidemiology; TB, tuberculosis; TS, treatment success.

(63.2% male, 36.8% female; mean age: 50.1) and 29,988 (56.5% male, 43.5% female; mean age: 54.7) patients. In both the private and public sectors, the proportion of patients with transfer history decreased (public: 23.8% to 21.7% and private: 20.8% to 17.6%), and bacteriological confirmed disease increased (public: 48.9% to 62.3% and private: 48.8% to 58.1%) in 2014 compared to 2009. After expanding nationwide PPM, absolute TS rates improved by 9.10% (87.5% to 93.4%) and by 13.6% (from 70.3% to 83.9%) in the public and private sectors. Relative to the public, the private saw 4.1% (95% confidence interval [CI] 2.9% to 5.3%, $p$-value < 0.001) and −8.7% (95% CI −9.7% to −7.7%, $p$-value <0.001) higher rates of improvement in TS and reduction in LTFU. Treatment outcomes did not improve in patients who experienced at least 1 transfer during their TB treatment. Study limitations include non-longitudinal nature of our original dataset, inability to assess the regional disparities, and verify PPM program's impact on TB mortality.

## Conclusions

We found that the nationwide scale-up of the PPM program was associated with improvements in TB treatment outcomes in the private sector in South Korea. Centralized financial governance and regulatory mechanisms were integral in facilitating the integration of highly diverse South Korean private sector into the national TB control program and scaling up of the PPM intervention nationwide. However, TB care gaps continued to exist for patients who transferred at least once during their treatment. These programmatic gaps may be improved through reducing administrative hurdles and making programmatic amendments that can help facilitate management TB patients between institutions and healthcare sectors, as well as across administrative regions.

## Author summary

### Why was this study done?

- With growing dominance of the private healthcare sector globally, majority of tuberculosis (TB) patients are seeking care in the private sector. Public–private mix (PPM) programs that can synergistically and comprehensively engage the private healthcare sector care into the national TB control efforts are integral in addressing TB care cascade disparities in the private sector.

- PPM interventions are widely being adopted by many high TB–burden countries with dominant private sectors, but current evidence is limited to small-scale pilot projects that were implemented in single or subregions less that did not sustain their operations for more than 2 years.

- To our knowledge, South Korea's PPM program is currently the only fully scaled up, long-standing nationwide program (formal inception in 2009). The South Korean experience can provide important insights on the long-term impact and policy gaps in integrating private sector TB care and patient management into the national TB control program.

## What did the researchers do and find?

- Using individual-level TB patient data registered in the Korean National Tuberculosis Surveillance System (KNTSS), we assembled cohort datasets classifying patients based on their primary institution reporting TB treatment registration (public versus private) in the respective years (2009 and 2014).

- Propensity score matching (PSM) using kernel method was performed to correct for imbalances in the observed covariates due to non-longitudinal and non-counterfactual nature in our data.

- Before the PPM program was formally implemented as a national program in 2009, private sectors institutions had suboptimal treatment outcomes compared to those in the public sector. Using difference in differences (DID) analysis, our study confirmed that the implementation and expansion of the nationwide PPM program was associated with a reduction in TB treatment outcomes gaps between the 2 healthcare sectors.

## What do these findings mean?

- To our knowledge, our study provides first long-term and comprehensive evidence of the policy level impact of the nationwide PPM program in reducing TB care disparities between the 2 healthcare sectors.

- Although significant improvements in TB treatment outcomes have been observed in the private sector, programmatic and policy adjustments may be needed to address TB care gaps experienced by TB patients transferring institutions during their treatment.

- Our study provides important evidence, insights, and future directions for countries that aims to strengthen and scale up PPM programs for TB alongside the efforts to introduce social health insurance. The South Korean experience highlights the importance of strengthening the financial governance and regulatory mechanisms in implementing and scaling up the PPM intervention to streamline national TB control efforts.

## Introduction

The global tuberculosis (TB) control efforts are faced with the perpetual problems of the "missing millions" of TB patients—one-third of estimated TB cases—who are left undiagnosed or without treatment [1–3]. These consequences of TB care cascade gaps are major obstacles in achieving the ambitious post-2015 global TB goals and Sustainable Development Goals Target 3.3 [4–7] and are most prominent in the private sector. In 2017, more than two-thirds of TB patients initially sought and remained in care in the private sector, but the private sector notification only accounted for 19% of the total global TB notification (equivalent to 12% of estimated TB incidence in 2017) [8]. This conflicting phenomenon signifies an urgent need for context-specific health systems strategies that can induce necessary behavioral changes to cohesively engage all private sectors in optimizing the care pathways for TB patients.

   The public–private mix (PPM) strategy has been central to increasing private sector engagement and leveraging overall TB control efforts globally since 2001 and is being supported

through a wide range of activities from development of generic guidelines and toolkits to formation of global PPM working groups hosted by the Stop TB Partnership [9]. Evidence from more than 48 pilot PPM programs in high TB–burden countries have demonstrated improvements in TB case detection, treatment adherence and outcomes, equity in access to TB care, and TB care costs [10,11] as direct results of PPM implementation. However, many of these programs did not fully engage all private sector stakeholders and struggled to scale up as nationwide programs due to weak enforcement of regulations and challenges in securing sustainable financing [10,12,13]. India has made significant progress in increasing TB notification and treatment success (TS) rates in the private sector by consolidating government efforts to scale up private provider engagement interventions through PPM (under the National Multisectoral Action Framework for TB-Free India); yet, it has faced a wide range of challenges in implementing, scaling up the coverage, and data reporting for its PPM program [14]. As such, there lacks long-term and comprehensive evidence generated from a fully scaled up national PPM program(s) [15] that can help facilitate further policy development for PPM program scale-up in countries aiming to improve TB control efforts in the private sector.

In this regard, the South Korean PPM program is currently the only fully scaled up nationwide program and can provide an important insights and road map for those countries making efforts to develop plans to scale up and sustain PPM as a nationwide program. The South Korean PPM is backed by the single-payer National Health Insurance (NHI) system, which regulates fee-for-service payments of all healthcare services provided in South Korea [16]. As such, patients can freely choose health providers and are only required to make standardized co-payments depending on the fee schedules for relevant health services utilized [17]. Operationally, PPM-dedicated nurses are central proponent of the South Korean PPM who provide individualized patient education and treatment adherence monitoring services to TB patients in the private sector (Fig 1A) [18,19]. Initially starting with 22 private sector institutions in 2009, the South Korean PPM program rapidly expanded to cover more than 120 private sector institutions as of 2017, and during these years, TS among smear positive TB patients managed in the private sector has improved to 88.3% from 68% [20].

Current published evidence documenting the successes of the South Korean PPM are largely limited to studies conducted at a subset of private hospitals participating in the PPM program [21]. Therefore, using individual patient-level notification data from the Korean National Tuberculosis Surveillance System (KNTSS), we investigated the impact of fully implemented nationwide PPM program on the Korean TB care cascade during its first 5 years of operation (2009 to 2014) with a focus on improvements in TS and loss to follow-up (LTFU). Furthermore, we also evaluated patient-level factors influencing the outcomes in TB treatment in the South Korean private sectors.

## Methods

### The PPM program in South Korea

In South Korea, the Korean Centers for Disease Control and Prevention (KCDC) is the governing organization of the National Tuberculosis Program (NTP), which manages all TB control programs in both the public and private health sectors [18]. The Korean Society of TB and Lung Diseases jointly develops clinical guidelines for TB care and management with the KCDC. All healthcare providers in both the public and private sectors are mandated to follow these guidelines and report to the NTP if patients are diagnosed and treated for TB in their institutions. In general, patients can freely decide where they choose to seek healthcare services [16], including TB-related services. In certain circumstances (e.g., patients seeking care at private primary clinics that are not specializing in respiratory illness or those facilities without TB

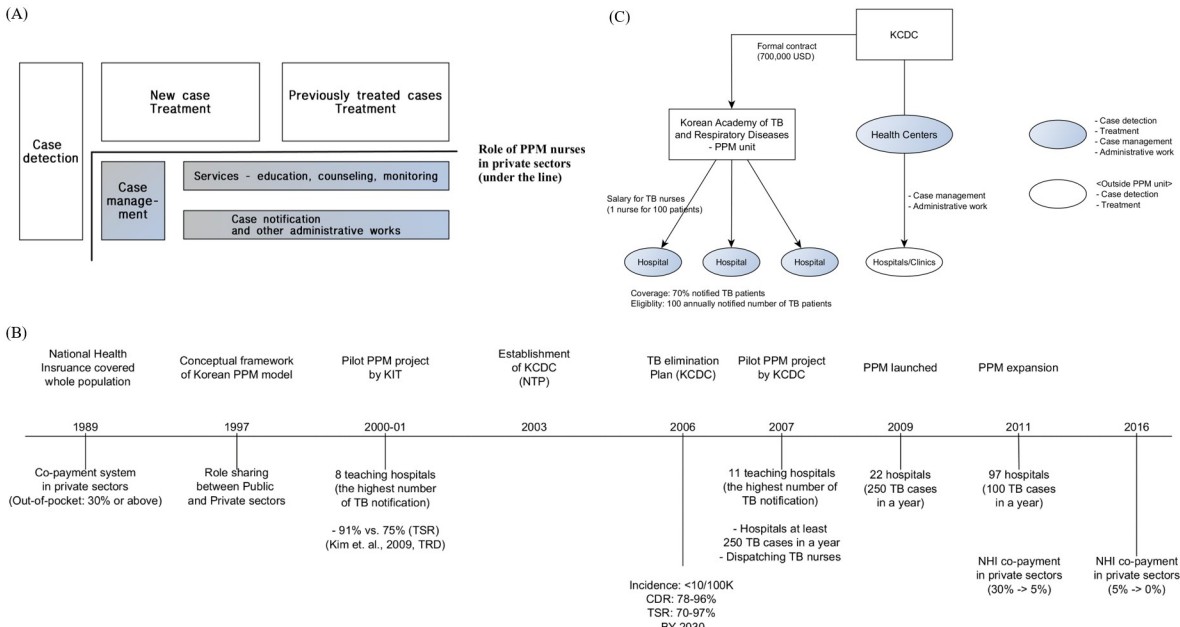

**Fig 1. Background of the TB program and PPM model in South Korea.** **(A)** Role of PPM nursed in private sectors. **(B)** Historical overview of the Korean PPM and the NHI. **(C)** Simplified illustration of the South Korean PPM program components. Panel (A) was adapted from Kim and colleagues and illustrates distribution of TB control and management roles in South Korea. Blue box denotes roles delegate to PPM nurses. Roles in white boxes are those delegated to healthcare institutions in both the private and public sectors. Panel (B) provides a historical overview of the South Korean PPM under NHI scheme. The NHI system completed full national coverage in 1989. In 1997, the first conceptual model for the South Korean PPM was developed and piloted in 2000. In 2003, KCDC was established, and first national TB elimination plan by KCDC was developed (2006), which PPM was one of its main components. In 2009, the nationwide PPM was first implemented in 22 hospitals reporting at least 250 TB cases annually and was rapidly expanded to 97 private sector hospitals by 2011 that reported at least 100 TB cases annually. In 2016, NHI co-payments for TB services in the private sector were eliminated to reduce financial burden for TB patients seeking care in the private sector (TB care in the public sector was free prior to date). Panel (C) provides a simplified schematic illustrating the South Korean PPM program components. KCDC, through public health centers and the Korean Academy of TB and Respiratory Disease, governs the PPM program PPM and awards salary contracts to PPM nurses who are hired directly by each PPM-affiliated hospitals (1 PPM nurse per 100 TB patients on average). As of 2019, 70% of all TB patients notified in South Korea are managed in the PPM-affiliated private healthcare institutions. KCDC, Korean Centers for Disease Control and Prevention; KIT, Korean Institute of Tuberculosis; NHI, National Health Insurance; NTP, National Tuberculosis Program; PPM, public–private mix; TB, tuberculosis; TB nurses, PPM nurses; TRD, tuberculosis respiratory disease; TSR, treatment success rate.

services), patients may be referred to higher level institutions (e.g., public or private tertiary hospitals or specialized TB hospitals) if determined to have symptoms suggestive of TB or diagnosed as TB. Decisions for hospitalization are solely based on patient's medical conditions (e.g., severity of disease symptoms and comorbidities) and relevant clinical guidelines and not by where patient sought care for their illness.

Since the Korean War (1950 to 1953), South Korea has made significant strides in tackling the TB problem alongside of the country's rapid economic development, resulting in improvements in the general health and socioeconomic status [22]. However, with more TB patients seeking care in the highly dispersed private sector, TB control successes in the public sector were undermined by large gaps in TB case notification, high transfer-out, and LTFU in the private sector [23]. Subsequently, post-1990s was marked by "a period of stagnation" in TB control with TB notification fluctuating between 85 and 100 per 100,000 [22].

To address the growing TB control disparities in the private sector, the KCDC established a PPM pilot program at 11 tertiary care private hospitals annually, reporting more than 250 TB patients in 2007 with 16 PPM-designated nurses (250 TB patients per PPM nurse) [24]. Backed by government's strong financial and regulatory commitment [25], the South Korean PPM

program was formally implemented as a national program in 2009. In subsequent years, PPM program was expanded to private healthcare institutions notifying at least 100 TB patients [26]. Along with the program expansion, the PPM nurse to TB patient ratio was further reduced (100 TB patients per PPM nurse) to improve workloads and facilitate TB patient management provided by PPM nurses [26]. As of 2019, more than 70% of all notified TB patients reported from 154 PPM-affiliated private institutions were managed by a total of 258 PPM nurses [27]. Formative periodic training for PPM nurses is provided by the Korean Institute of Tuberculosis [24]. A brief historical overview of the South Korean PPM program is illustrated in Fig 1B.

As with the earlier pilot PPM models [28], the current nationwide program was developed around dedicated PPM nurses who are central enactors in ensuring private sector adherence to the national TB control guidelines (Fig 1C). Main duties of PPM nurses include (1) registering TB patients and regularly updating patient information in the KNTSS; (2) educating TB patients (contents include informing patients on the importance of TB treatment and adherence to the treatment schedule, risks of development of drug resistance, basic knowledge of TB disease, a need for contact investigation for household members and frequent contacts of the patients, and potential treatment side effects and how to cope/manage these side effects); (3) phone call reminders prior to patient's treatment follow-up appointments; (4) checking patient's TB treatment adherence and anti-TB drug adverse events; and (5) follow-up calls for those patients missing their clinic visit appointments [21].

## Study design and data preparation

In order to evaluate the policy impact of the PPM program on key patient outcome measures (TS and LTFU), we assembled cohort datasets of registered TB patients in the years 2009 and 2014, using TB notification data from the KNTSS managed by the KCDC. Our study was conceptualized in 2016, and the data were extracted from the KNTSS database in 2017. Complete TB notification data were extracted up to year 2016; however, treatment outcome data for those patients started TB treatments in 2015 and 2016 were incomplete at the time of the data extraction. Therefore, we focused our assessment on TB patients registered in 2014 as a comparator cohort to those in 2009, the year in which the nationwide PPM program was launched. Each TB patient in the respective cohort years was categorized either as private or public sector patient based on the primary institution reporting and updating patient information to the NTP.

A total of 33,591 and 32,791 individual patient records were identified for 2009 and 2014 cohorts based on our inclusion criteria (Fig 2A and 2B). We only included newly identified (coded in the KNTSS as "new cases") TB patients who were not previously registered in the KNTSS since 2001. Previously registered patients who had treatment duration less than 30 days were also classified as new cases. We excluded patients for whom we could not properly assess their treatment duration (e.g., there were several cases whose treatment completion dates were logged as dates prior to the treatment initiation date), who had changes in their diagnosis such as non-tuberculosis mycobacterium, or had drug-resistant (DR) TB (96 and 538 patients in 2009 and 2014 cohorts: represents 0.3% and 1.6% of all patients in the original KNTSS data used for this study). We further removed 53 patients (0.2% of the cohort, all in the 2009 cohort) with missing covariate (3 cases had independent variable missing, and the rest had age or/and sex variables missing) values.

The private sectors included university hospitals, private secondary and tertiary hospitals, and primary private clinics, while the public sector included public health centers and national hospitals. For each individual patient included in the respective cohorts, we retrieved data on

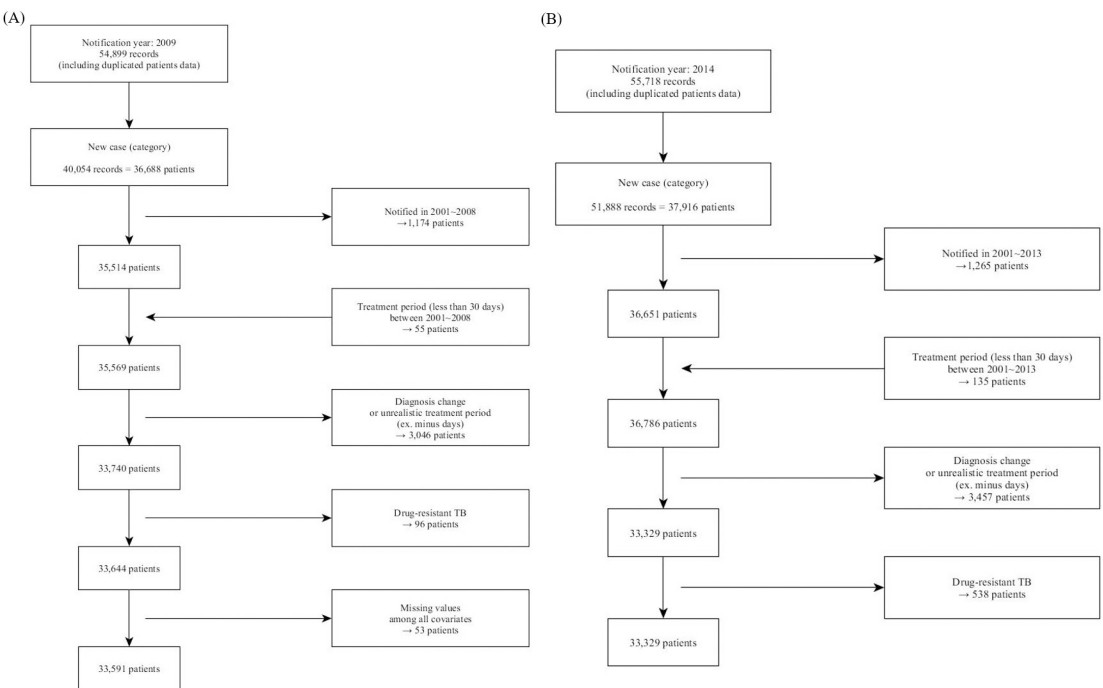

**Fig 2. Flow diagrams of the study population in (A) 2009 and (B) 2014.** TB, tuberculosis.

sex, age, nationality, transfer history, geography, and diagnostic test results (chest X-ray [CXR], smear, and culture). Age was categorized by 10-year increments for those 20 years and older and less than 70 years. Patients who were younger than 20 years or older than 70 years (inclusive) were classified into single age groups of "0 to 19 years" and "70 or older." Patient's nationality was coded as "Korean" for those who had the Korean citizenship at the time of their TB diagnosis. All other patients were classified as "non-Korean citizens." Patient's transfer history was also coded as a binary variable, distinguishing between those managed exclusively at 1 institution and those who have had at least 1 transfer history to another institution. Patient's geographic residence information was coded based on the national administrative division and unit classification—metropolitan, city, and town—assigned based on patient's address reported (pre-classified in the KNTSS and blinded to the investigators) in the national identity registration database. Each patient's TB diagnosis was classified based on 3 types of diagnostic test results: CXR, smear microscopy, and culture. CXR results were classified as those with "findings suggestive of TB," "normal," or "other disease." Smear microscopy and culture results were used to designate each patient's status of bacteriologic diagnosis where these test results were classified as "positive," "negative," or "unknown." According to all diagnostic results, bacteriological status was classified based on the International Classification of Diseases-10th Revision (ICD-10) code recorded in the KNTSS database (A15.X: bacteriologically confirmed, A16.X: not confirmed, and others: extrapulmonary [EP]).

Patients' treatment outcomes were categorized as cure, completion, failure, death, or LTFU in the KNTSS database. Patients were defined as LTFU if they had unknown treatment results (marked in the database as "not evaluated") or experienced treatment interruption (marked as "interrupted"), calculated as a proportion (the sum of patients with unknown treatment results and interrupted treatment over a total number of new TB patients registered on KNTSS within each designated cohort year). Treatment interruption was classified for those patients who did

not have treatment follow-up record of more than 3 months since their last treatment follow-up visit date. Likewise, those patients who were transferred to another institution (within and across the 2 health service sectors) with more than 3 months of treatment data missing post transfer date were also marked as LTFU. For patients with multiple registration (as new TB patient) in the same year, we considered patients were LTFU if the time between the 2 registration dates were less than 3 months (Fig A in S1 File). LTFU was calculated as the proportion of total number of LTFU patients over total number of new TB patients registered in the KNTSS for each designated cohort year. TS was calculated as the sum of all patients "cured" and "completion" over the total number of new TB patients registered on KNTSS in each cohort year. TB patients with at least 5 full months of treatment record without any evidence of treatment failure were also assessed as "treatment completed."

## Statistical analysis

Comparison of the baseline covariates—age, sex, nationality, transfer history (treatment reported at 1 or more institutions since reported as a TB case), geographic division (categorized as metropolitan, city, and towns designated), diagnostic test results (CXR, smear, and culture)—were assessed using the chi-squared test.

To assess the causal effect of the Korean PPM program on TB treatment outcomes, we used the difference in differences (DID) analysis. In the field of TB, DID analysis has previously been used to evaluate long-term effects of pre-1950s TB policies/public health campaigns (limited to the United States of America and Denmark) using historical data [29–31]. We compared the changes in treatment outcome (TS and LTFU) between the public and private sector TB patient cohorts in 2009 and 2014. Therefore, our null hypothesis assumed that, in absence of the PPM intervention (policy), there will be no difference in treatment outcome improvements between the 2 sectors. For our primary analysis, we performed propensity score matching (PSM) analysis to adjust for imbalances in the covariates between the 2 population cohorts due to the non-longitudinal and non-counterfactual nature of our analysis (Fig C in S1 File). We used the kernel method—nonparametric matching estimator—to construct the counterfactual private sector patient cohort matched based on weighted averages of key patient-level parameters (age, sex, nationality, transfer history, and diagnostic test results) of public sector patients. For DID analysis, the kernel PSM method is preferred over other PSM methods (e.g., 1:n matching) as it can achieve lower variance by using more information available in the data [32].

The basic equation for the DID analysis was constructed as $Y = \beta_0 + \beta_1 T + \beta_2 I + \beta_3 TI + \gamma X + e$, with $Y$ defined as outcome measure, coded as a binary response variable (LTFU: 0 = others, 1 = LTFU or unknown and TS: 0 = others, 1 = cured or completed), $T$ as time variable (2009 = 0 and 2014 = 1), $I$ as the intervention variable (public = 0 and private = 1), and X as a vector of covariates on age, sex, nationality, and test results (CXR, smear, and culture tests). The estimated coefficients have the following interpretation: $\widehat{\beta}_0$ is the mean outcome for the public sector (control) in 2009. $\widehat{\beta}_0 + \widehat{\beta}_1$ is the mean outcome for the public sector (control) in 2014. $\widehat{\beta}_2$ is the single difference between private (intervention) and public (control) groups in 2009. $\widehat{\beta}_3$ is the DID estimate quantifying absolute differences in TB treatment outcomes between 2009 and 2014 in patients managed in the private sector compared to those in the public sector as a result of the nationwide PPM intervention. To test the robustness of our analyses and conclusions, we compared PSM DID estimates with that of the crude (original data) DID estimates as well as other matching methods used for PSM (1:1 and 1:3 matching). Outputs from these analyses are available in our supplement (Table A in S1 File). Data curation and analyses were performed using Stata v15.0 (Stata, College Station, Texas, USA). This

study is reported according to the Strengthening the Reporting of Observational Studies in Epidemiology (STROBE) guideline (S2 File).

## Ethics committee approval

Our study protocol (S3 File) was reviewed by the Institutional Review Board at the Korean National Tuberculosis Association (2018-KNTA-IRB-006), and our study was granted ethics exemption as data on patient identifiers were removed and/or provided as encrypted data from the KCDC.

# Results

## Characteristics of study population

Demographic and clinical characteristics of 66,382 TB patients in our study are summarized in Table 1. Between the 2 cohort years, the number of TB patients managed in the public sector decreased ($n$ = 6,195 (18.4%) in 2009 to $n$ = 2,803 (8.6%) in 2014), while there was an increase in the number of TB patients in the private sector ($n$ = 27,396 (81.6%) in 2009 to $n$ = 29,988 (91.5%) in 2014). The proportion of the patients who were non-Korean citizen was higher in the public sector in both years (5.8% and 8.9% compared to 3.8% in the private sector in 2009 and 2014). Between 2009 and 2014, the proportion of bacteriologically confirmed TB patients increased (public: 48.9% to 62.3% and private: 48.8% to 58.1%). Additionally, TB patients transferring to another institution during their treatment (at least once or more) decreased (public: 23.8% to 21.7% and private: 20.8% to 17.6%).

## Impact of PPM on treatment outcomes

During the first 5 years of the scale-up of the of formal nationwide PPM program, TS improved from 87.5% to 93.4% (absolute improvement of 9.10%) and 70.3% to 83.9% (absolute improvement of 13.6%) in the public and private sectors, respectively (Table 2). Although TS was consistently lower in the private sector, a positive DID estimate of 0.041 (or 4.1%; 95% confidence interval [CI], 2.9 to 5.3%. $p$-value < 0.001)—calculated based on the differences of the absolute improvements in TS rate in the respective sector (13.3% to 9.10%)—indicates that the rate of improvement was higher (or impact of the scale-up of the nationwide PPM was greater) in the private sector (Fig 3B, red lines) relative to the public sector.

For LTFU, the private sector had significantly higher proportion of LTFU (25.2%) in the baseline year relative to the public sector (11.3%) in the inception year of the nationwide PPM (Table 2). In 2014, the proportion of TB patients LTFU out of all patients enrolled in TB treatment fell to 7.3% and 5.3% in the private and public sectors, respectively, achieving an absolute decrease in LTFU of 2.0% and 10.7% for the respective sectors. A negative DID estimate of −0.087 (or −8.7%; 95% CI −9.7% to −7.7%, $p$-value < 0.001)—calculated based on the differences of the absolute decrease in LTFU (−2.0% to 10.7%)—indicates the rate of decline in LTFU was faster in the private sector (Fig 3B, blue lines) compared to the public sector as a result of the PPM expansion.

A summary of the PSM DID estimates for TS and LTFU are presented graphically in Fig 4 and in Tables B and C in S1 File. For certain patient subgroups (60 years or older, those with transfer history, and positive culture results), DID estimate was not statistically significant (i.e., 95% CI crossing 0.0), inferring that the PPM program may not have made clear impact on improving TS in these subgroup populations. Particularly, if patients had a transfer history during their TB treatment (Fig 4), treatment outcomes worsened. In this patient group of the private sector, TS declined by 9.2% (95% CI −6.3% to −12.1%, $p$-value <0.001), while LTUF

**Table 1. Baseline characteristics of study population by treatment institutions and cohort years (n = 66,382).**

| | 2009 (n = 33,591) | | | 2014 (n = 32,791) | | |
|---|---|---|---|---|---|---|
| | **Public** | **Private** | **p-value** | **Public** | **Private** | **p-value** |
| | **N (%)** | **N (%)** | | **N (%)** | **N (%)** | |
| Total | 6,195 (18.4) | 27,396 (81.6) | | 2,803 (8.6) | 29,988 (91.5) | |
| Sex | | | <0.001 | | | <0.001 |
| Male | 3,903 (63.0) | 15,365 (56.1) | | 1,771 (63.2) | 16,942 (56.5) | |
| Female | 2,292 (37.0) | 12,031 (43.9) | | 1,032 (36.8) | 13,046 (43.5) | |
| Age, years[1] | | | <0.001 | | | <0.001 |
| 0–19 | 923 (14.9) | 1,470 (5.4) | | 253 (9.0) | 990 (3.3) | |
| 20–29 | 1,234 (19.9) | 4,203 (15.3) | | 527 (18.8) | 3,431 (11.4) | |
| 30–39 | 980 (15.8) | 3,938 (14.4) | | 361 (12.9) | 3,398 (11.3) | |
| 40–49 | 948 (15.3) | 4,121 (15.0) | | 447 (15.9) | 4,243 (14.2) | |
| 50–59 | 746 (12.0) | 3,786 (13.8) | | 489 (17.5) | 5,046 (16.8) | |
| 60–69 | 550 (8.9) | 3,671 (13.4) | | 291 (10.4) | 3,832 (12.8) | |
| ≥70 | 814 (13.1) | 6,207 (22.7) | | 435 (15.5) | 9,048 (30.2) | |
| Nationality[2] | | | <0.001 | | | <0.001 |
| Korean | 5,833 (94.2) | 26,342 (96.2) | | 2,552 (91.1) | 28,844 (96.2) | |
| Non-Korean citizen | 362 (5.8) | 1,054 (3.8) | | 251 (8.9) | 1,144 (3.8) | |
| Transfer history[1] | | | <0.001 | | | <0.001 |
| Treated at 1 site | 4,722 (76.2) | 21,705 (79.2) | | 2,194 (78.3) | 24,718 (82.4) | |
| More than 1 site | 1,473 (23.8) | 5,691 (20.8) | | 609 (21.7) | 5,270 (17.6) | |
| Geography[1] | | | <0.001 | | | 0.015 |
| Metropolitan | 3,028 (48.9) | 15,858 (57.9) | | 1,314 (46.9) | 13,211 (44.1) | |
| City | 2,561 (41.3) | 10,801 (39.4) | | 1,184 (42.2) | 13,412 (44.7) | |
| Town | 606 (9.8) | 737 (2.7) | | 305 (10.9) | 3,365 (11.2) | |
| CXR[1] | | | <0.001 | | | <0.001 |
| TB signs | 5,136 (82.9) | 21,986 (80.3) | | 2,624 (93.6) | 23,156 (77.2) | |
| Normal | 58 (0.9) | 1,015 (3.7) | | 52 (1.9) | 2,500 (8.3) | |
| Other disease | 1,001 (16.2) | 4,395 (16.0) | | 127 (4.5) | 4,332 (14.5) | |
| Bacteriological status[1] | | | <0.001 | | | <0.001 |
| Confirmed | 3,029 (48.9) | 13,366 (48.8) | | 1,747 (62.3) | 17,418 (58.1) | |
| Not confirmed | 3,142 (50.7) | 11,287 (41.2) | | 1,050 (37.5) | 8,422 (28.1) | |
| EP | 24 (0.4) | 2,743 (10.0) | | 6 (0.2) | 4,148 (13.8) | |
| Smear[1] | | | <0.001 | | | <0.001 |
| Positive | 2,313 (37.3) | 8,616 (31.4) | | 1,116 (39.8) | 8,498 (28.3) | |
| Negative | 3,694 (59.6) | 10,571 (38.6) | | 1,637 (58.4) | 15,867 (52.9) | |
| Unknown | 188 (3.0) | 8,209 (30.0) | | 50 (1.8) | 5,623 (18.8) | |
| Culture[1] | | | <0.001 | | | <0.001 |
| Positive | 2,324 (37.5) | 3,984 (14.5) | | 1,646 (58.7) | 13,619 (45.4) | |
| Negative | 2,417 (39.0) | 2,765 (10.1) | | 997 (35.6) | 8,461 (28.2) | |
| Unknown | 1,454 (23.5) | 20,647 (75.4) | | 160 (5.7) | 7,908 (26.4) | |

[1]The difference of proportions significantly in the overall, public, and private comparison between 2009 and 2014.

[2]The difference of proportion significantly in the public comparison between 2009 and 2014.

CXR, chest X-ray; EP, extrapulmonary; TB, tuberculosis.

**Table 2. Treatment outcomes of public and private sectors in 2009 and 2014 (%).**

|  | 2009 | | 2014 | |
|---|---|---|---|---|
|  | **Public** | **Private** | **Public** | **Private** |
| TS | 87.46% | 70.34% | 93.36% | 83.86% |
| Failure | 0.50% | 0.11% | 0.07% | 0.07% |
| LTFU | 11.32% | 25.20% | 5.32% | 7.32% |
| Deaths | 0.73% | 4.35% | 1.25% | 8.75% |

LTFU, loss to follow-up; TS, treatment success.

increased by 6.7% (95% CI 4.0% to 9.4%, *p*-value <0.001) relative to the public sector patients with transfer history between to the cohort periods.

There was no difference in the direction of the DID estimates with and without PSM (Table D in S1 File), further demonstrating that the PPM intervention was associated with improvements in TB treatment outcomes in the private sector relative to the public sector (Table E in S1 File). However, the crude DID estimates for certain patient groups indicated that there was no effect of PPM on TS (patients with normal CXR results and EP-TB) and LTFU (patients with EP-TB and unknown smear status), which contradicted the PSM DID estimates.

## Discussion

To our knowledge, our study is first to comprehensively evaluate the long-term policy impact of the South Korean PPM program using patient-level data. In this study, we established that the nationwide scale-up of the PPM program was associated with an improvement in patient treatment outcomes in the South Korean private sector. Using the DID analytic framework, we showed that gaps in TS and LTFU between the private and public sector patients were likely to be reduced after expanding the nationwide PPM intervention between 2009 and 2014. Improvements in treatment outcomes were consistently observed in all demographic, socio-economic, and clinical status strata for those patients managed in the private sector compared to the public sector TB patients. However, we noted that several patient subgroups, particularly those patients with a history of transferring to another institution during their TB treatment, did not benefit from the nationwide PPM intervention. This highlights an important area which future iterations of PPM policy and programmatic implementation should address as the South Korea aims to further strengthen its TB control program in the private sector through PPM intervention.

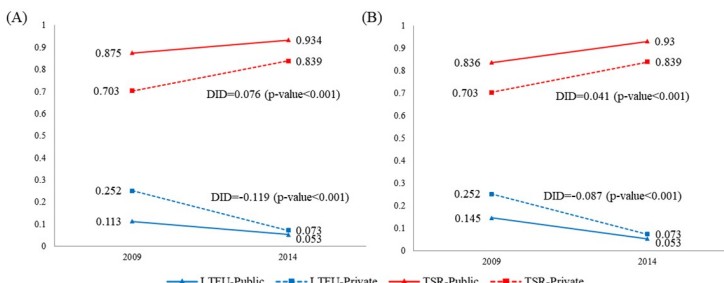

**Fig 3. DID on TS and LTFU in public and private sectors in 2009 and 2014. (A)** Crude DID estimate. **(B)** DID estimate with PSM. DID, difference in differences; LTFU, loss to follow-up; PSM, propensity score matching; TS, treatment success; TSR, treatment success rate.

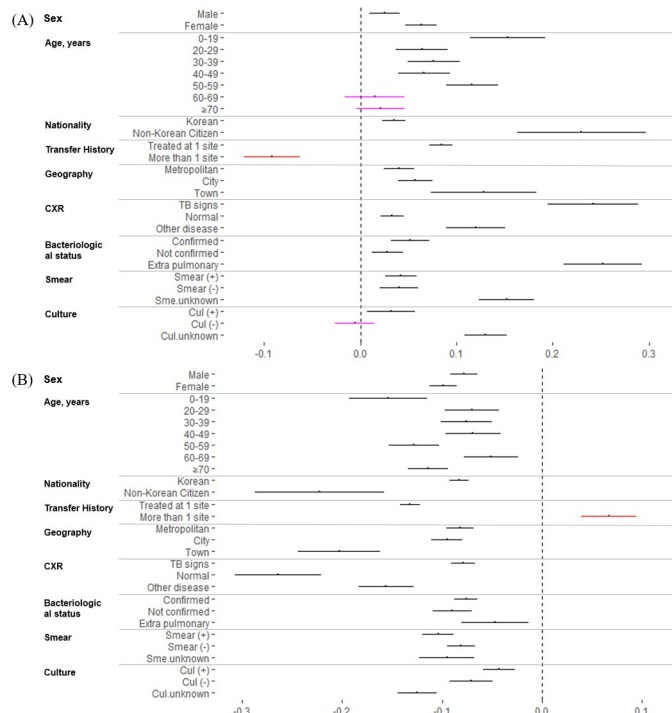

**Fig 4. Subgroup analysis of DID with PSM on TS and LTFU in public and private sectors in 2009 and 2014. (A)** Patient subgroup DID estimates for TS. **(B)** Patient subgroup DID estimates for LTFU. Panels (A) and (B) present d estimates for TS and LTFU by each patient subgroups. For each patient subgroup, size of the lines represents the 95% CIs with the primary DID estimate shown as an indented dot at the middle of each line. Patient subgroups with red colored lines represents groups that were negatively impacted by the nationwide PPM (with the primary DID and 95% CIs falling below or above "zero." Pink colored lines are assigned when 95% CIs of DID estimates cross the "null," which suggest that the nationwide PPM did not have an effect in improving TB treatment outcomes. CI, confidence interval; Cul, culture; CXR, chest X-ray; DID, difference in differences; LTFU, loss to follow-up; PSM, propensity score matching; Sme, smear; TB, tuberculosis; TS, treatment success.

It is important to note that the foundation of sustainable growth and successes of the South Korea's PPM program is anchored on the NHI scheme. First established as a mandate for large businesses to provide health insurance for their employees in 1977, the South Korean NHI scheme reached full national coverage by 1989 and improved coverages for health services provided in the private sector [16]. The single-payer NHI scheme allows for centralized financial governance of health services in South Korea, which simplified the financing mechanisms for implementing and scaling up the PPM program in highly privatize South Korean healthcare system (the private sector supplies more than 90% of all health services) [16]. Furthermore, alongside PPM program expansion, the NHI expanded its fee-for-service coverage for all TB-related health service as part of the comprehensive national strategic plan for TB control, which aimed to reduce both financial and health systems barriers for patients suffering from TB illness [18,33,34]. Subsequently, as of July 2016, TB patient seeking care in the private sector are exempt from co-payment when receiving TB-related health services. Likewise, these concurrent regulatory developments and sustainable financing mechanisms likely created important synergy with the development and expansion of the South Korea PPM in reducing disparities in TB treatment outcomes between the public and private healthcare sectors. Similarly, designing and optimizing scale-up plans for PPM programs around various ongoing efforts to improve social health insurance coverage (one of several ways to achieve Universal Health Coverage) in high TB–burden settings [35], it will be critical to design and plan PPM

program and its expansion optimized around the development and growth of the UHC to address increasing TB care disparities as the private sector increases its diversity and market share for medical care.

As with any large scale and comprehensive public health interventions, our study identified several policy gaps during the first 5 years of the nationwide PPM program in South Korea. First, there was a clear trend of diminishing impact of PPM with certain subgroups: The impact of the PPM program for elderly (60 or older) and culture confirmed TB patients managed in the private sector may not have been as beneficial as the same patient groups managed in the public sector. More importantly, for patients with a transfer history—close to a quarter of patients in the private sector—treatment outcomes worsened during the 5-year period (Fig 4A and 4B). This is likely attributable to administrative and programmatic challenges in managing TB patients when patients transfer institutions across and within health sectors (e.g., private to public and vice versa) and between different administrative regions. Likewise, facilitation of administrative processes and shared responsibility in managing TB patients across health sectors should be prioritized for future policy and programmatic updates for the South Korean PPM. In addition, we identified that non-Korean citizens were more likely to seek TB care in public sectors than those with the Korean citizenship. While this may be related to the issues of socioeconomic and cultural barriers faced by non-Korean citizens who may also fall outside of the NHI coverage (e.g., those foreigners who do not hold proper legal status in Korea), we cannot provide clear inference on this matter due to limitations in our data. Given the growing number of TB burden among the immigrants and foreign workers in South Korea [36], future studies should investigate gaps in TB care cascade and care-seeking barriers for the vulnerable immigrant population in South Korea.

There are several limitations that should be noted when interpreting our study findings. First, our study is not a longitudinal analysis of the same population observed over the study period, which are generally used for traditional DID analyses. As treatments for active TB disease generally do not last more than 6 months (in the case of DR TB, treatment may last 2 or more years, but our study focused on drug-susceptible TB patients), public and private sector TB patients in our 2009 and 2014 cohorts are constituted of completely different individuals (i.e., covariate measures are not repeated measures from the same TB patient). To address this problem, we used PSM to enhance comparability between the public and private as well as 2009 and 2014 cohorts in evaluating the impact of the PPM on treatment outcomes in TB patients (results showing the balance of covariates between the comparator groups are shown in Fig C in S1 File).

Second, it can be inferred that the South Korean PPM program may not have had an equal level of impact in improving TB deaths as compared to TS and LTFU. While death is an important outcome indicator to evaluate the impact of PPM interventions, estimates for TB deaths are likely influenced by factors such as age and severity of disease. As such, we considered TS and LTFU to be more appropriate and direct indicators to assess the policy level impact and primary function of the PPM (i.e., improving patient's treatment management and adherence). In South Korea, TB disease burden is higher in the elderly population (can naturally lead to more observation of death regardless of the PPM intervention; Table 2). Moreover, there was a steady decline of TB patients managed in the public sector over the 5-year period. With the complete NHI service coverage of all TB-related health services provided in the private sector since 2011, it is likely that patients preferred to be treated in the private sector where service and infrastructure are more comprehensive and superior to those provided in the public sector. Similarly, the private sector generally manages more serious cases of TB. Consequently, it can be inferred that public sector patients may be suffering from less complicated disease and thus may naturally have a better chance for successful treatment outcomes

with minimal intervention. Unfortunately, both of these factors (patient's socioeconomic position, care-seeking preferences, and disease severity) could not be assessed through data that were available for our study. Third, following caution should be taken in interpreting our results for generalization. Causal inference made based on DID method requires that there is no difference in trends in the outcomes between the compared groups (parallel trend assumption) and adjustment for clustering effects [37,38]. Prior to nationwide implementation of the PPM program in South Korea, we verified that treatment outcome trends in the public and private sectors were similar in that both sectors improved in parallel to one another (Fig B in S1 File). Due to limitations in our data, we were not able to properly investigate effects of clustering in uncertainty ranges of our DID estimates. While effects of clustering cannot be fully ignored, considering (1) rapid nationwide scale-up PPM intervention; and (2) central policy administrative governance of TB-services by KCDC and financial governance of the entire healthcare services through NHI system, there are likely minimal differences in policy administration and TB-related service provision regardless of where patients seek care (except in institutions without PPM nurses). Nonetheless, future studies investigating effects of nationwide policies (or interventions that involve large number of clusters or expecting variable effects across different clusters) should carefully consider effects of clustering in their analyses [37–39].

## Conclusions

Our DID analysis using patient-level data demonstrates that the development and expansion of the formal national PPM program was associated with significant long-term improvement in TB treatment outcomes in a highly privatized South Korean health system. However, our study also found important gaps in the PPM program, particularly for those patients with a history of transfers during their TB treatment. Likewise, strengthening TB patient management that can help facilitate the administrative process between institutions in both the public and private sectors and across PPM program's administrative regions may be needed to see further improvements in TB treatment outcomes in those patients transferring during their TB treatment. For countries currently developing initial and scale-up strategies for PPM as a national program, our study may provide important insights and future directions in synchronizing TB control efforts across the two healthcare sectors. For these countries, strengthening the financial governance and regulatory mechanisms to integrate key private sector providers in the national TB control program may be an important priority in effectively introducing the PPM as a nationwide program.

## Supporting information

**S1 File. Supporting information figures and tables.** Fig A: Treatment outcome assessment criteria for patients with multiple registration records in a given year. Fig B: A trend of TS rate between public and private sectors in 1987, 1993, 2009, and 2014. Fig C: Distribution of propensity scores before and after kernel matching in 2009 and 2014. Table A: Compared analysis of DID with various methods of PSM. Table B: Subgroup analysis of DID with PSM on LTFU rate in public and private sectors in 2009 and 2014. Table C: Subgroup analysis of DID with PSM on TS rate in public and private sectors in 2009 and 2014. Table D: Comparing the results of DID after using categorical and continuous variable. Table E: Absolute and relative risk differences between public and private sectors in 2009 and 2014. DID, difference in differences; LTFU, loss to follow-up; PSM, propensity score matching; TS, treatment success.
(DOCX)

**S2 File. STROBE Checklist.** STROBE, Strengthening the Reporting of Observational Studies in Epidemiology.
(DOC)

**S3 File. Research protocol.**
(DOCX)

**S4 File. TB case notification.** TB, tuberculosis.
(DOCX)

## Author Contributions

**Conceptualization:** Hyunwoo Kim, Hee-Jin Kim, Hongjo Choi.

**Data curation:** Hojoon Sohn, Hae-Young Kim.

**Formal analysis:** Sarah Yu, Hae-Young Kim, Hongjo Choi.

**Funding acquisition:** Hongjo Choi.

**Investigation:** Hyunwoo Kim, Hongjo Choi.

**Methodology:** Sarah Yu.

**Project administration:** Hongjo Choi.

**Resources:** Hyunwoo Kim.

**Software:** Sarah Yu.

**Supervision:** Hongjo Choi.

**Validation:** Hojoon Sohn, Hae-Young Kim, Kyung-Hyun Oh, Hee-Jin Kim, Haejoo Chung, Hongjo Choi.

**Visualization:** Sarah Yu.

**Writing – original draft:** Sarah Yu, Hojoon Sohn, Hongjo Choi.

**Writing – review & editing:** Sarah Yu, Hojoon Sohn, Kyung-Hyun Oh, Hee-Jin Kim, Haejoo Chung, Hongjo Choi.

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
