## [Decision Letter · Decision Letter 0]

9 Dec 2019

Dear Dr. Choi,

Thank you very much for submitting your manuscript "Impact Evaluation of Nationwide Public-Private Mix (PPM) Program for Tuberculosis Under National Health Insurance: A Difference-In-Difference Analysis of the South Korean PPM" (PMEDICINE-D-19-03769) for consideration at PLOS Medicine. 

[LINK]

In light of these reviews, I am afraid that we will not be able to accept the manuscript for publication in the journal in its current form, but we would like to consider a revised version that addresses the reviewers' and editors' comments. Obviously we cannot make any decision about publication until we have seen the revised manuscript and your response, and we plan to seek re-review by one or more of the reviewers. 

We expect to receive your revised manuscript by Dec 30 2019 11:59PM. Please email us (plosmedicine@plos.org) if you have any questions or concerns.

We look forward to receiving your revised manuscript. 

Sincerely,

Thomas McBride, PhD

Senior Editor 

PLOS Medicine

plosmedicine.org

Abstract- I think there are too many acronyms in the abstract and would suggest removing some to clean it up – DID, TSR and LTFU. Also please provide a sentence or two on the private and public health care in South Korea, ratio of patients registered to either for example and benefits of one over the other or bottlenecks. Please add p values with 95% Cis and add a sentence of the study’s limitations as the final sentence of the ‘Methods and Findings’ section, please. There appears to be a lack of information regarding the number of TB patients in the abstract, such as the number of patients followed and some summary demographic information. Please provide. Conclusion section of the abstract – please avoid language such as ‘clear evidence’ as this is not a trial. Please tone down such language. 

Data – please provide the URL for the KCDC for authors who meet requirements.

At this stage, we ask that you include a short, non-technical Author Summary of your research to make findings accessible to a wide audience that includes both scientists and non-scientists. The Author Summary should immediately follow the Abstract in your revised manuscript. This text is subject to editorial change and should be distinct from the scientific abstract. Please

see our author guidelines for more information: https://journals.plos.org/plosmedicine/s/revising-your-manuscript#loc-author-summary

References in the main text – please use square brackets instead of superscript, per PLOS style. 

Can you please comment on why the data only runs to 2014? It’s quite dated, as such

Can you please ensure any questionnaires are included as Supp files, for example any used by the nurses in education or phone calls (and translated, as necessary).

You say that anyone who isn’t Korean is referred to as foreign. I wonder if it’s possible to categorise in some way, say European, North America and so on…..? Also is it possible to comment on why foreigners access more public care?

Please add p values throughout, where appropriate. 

Line 303 – please remove ‘substantive’

Line 304 –‘ clear evidence’ please tone down language to something like, indicates …..may be important (instead of is essential)

Please do provide a reporting checklist – STROBE is probably the most appropriate. 

Did your study have a prospective protocol or analysis plan? Please state this (either way) early in the Methods section.

c) In either case, changes in the analysis—including those made in response to peer review comments—should be identified as such in the Methods section of the paper, with rationale.

Comments from the reviewers:

Reviewer #1: See attachment

Michael Dewey

Reviewer #2: The publication by Choi et al. presents an evaluation of the public-private partnership model in South Korea and is interesting for several reasons:

i) the model is unique in that it capitalises on the universal health coverage

ii) South Korea has seen a rapid decline of TB incidence (a situation other countries will hopefully face in the next decade) - however compared to other high resource settings, TB incidence is still relatively high

iii) the analysis is innovative using a difference in difference methods and propensity score

Different models of public-private partnership for TB control in different setting rightly deserve focused attention and would be well suited to be published in Plos Medicine. 

As a general note, the manuscript would benefit from editing by a native (scientific) English speaker. I am myself not a native speaker, thus I have limited my comments on grammar and language to those paragraphs which could be considerably shortened or need clarifications. 

Please find my detailed comments below:

Introduction

Line 71: National Health Insurance Scheme - this needs to be explained in more detail, otherwise the reader will not be able to follow the discussion. 

Line 74-76: This sentence is extremely difficult to understand especially without a background of how the Korean TB program looks like. 

Methods

General comment: 

In order to understand the PPM and put it into context - it would be important to describe the South Korean TB programme. Where are patients diagnosed - in hospitals or outpatients? Are there dedicated TB clinics (i.e. is it a vertical program) or are TB patients seen in chest clinics. How is treatment managed programmatically - are most patients initiated in a hospital or as outpatients. If patients are initiate in hospital, how long do they stay in hospital and where do they received their outpatient care (at their GP, in an outpatient TB clinic affiliated to the hospital?

From a methodological point of view my main question is - why did the authors decide to use 2 cohorts rather than looking at the trends longitudinally - i.e. 2009-2014 spanning the period pre-implementation, scale-up and post implementation. 

Paragraph: "Public private mix in South Korea"

I would recommend to restructure this sexction, as it is difficult to follow. For example the second paragraphs describes the 2007 pilot and the 2011 nationwide roll-out of PPM, while the third paragraph mentions a much earlier pilot in 2001. The authors may want to include a figure describing timeline both with regards to the national health insurance scheme and the PPM

Line 92: consider replacing "rapid growth in TB patient" with maybe "more TB patients seeing care.."

Line 100: who pays the PPM nurses?

Line 108-109: "Modeled based on.." please edit - this might be "Models based on

Line 121-123: "Private or public sector designation for each TB patient in the respective cohort years was assessed based the primary institution reporting registration of patient's TB." Please edit for example "Each TB patients was categorised as receiving private or public sector care based on the designation of the institution registering the TB patient in with the NTP"

Line 125-126: "We included only the newly identified (coded in the KNTSS as 'new cases') TB patients who started their TB treatment in the respective cohort years who were not previously registered in the system in years before the designated cohort years (notified in 2001 ~ 2008 for 2009 cohort and 2001 ~ 2013 for 2014 cohort). Please edit "…TB patients who started TB treatment and were not previously registered for TB (since 2001)"

Line 130-131: "We excluded patients with inappropriate treatment period (cases that completed treatment before treatment initiating date)…" I am not sure I understand this part of the sentence. Does that mean a patient was registered and their date of completing treatment was incorrectly entered as being before treatment initiation? 

Line 143: - please include "variable" after binary.

Line 149-152: "Patient's examination for the TB diagnosis was classified based on three types of diagnostic test results. Chest X-ray (CXR) results were classified as those with "presumptive TB", "normal", or "other disease". Bacteriological status was classified based on the ICD-10 code recorded in the KNTSS database (A15.X:

152 bacteriologically confirmed, A16.X: not confirmed, others: extrapulmonary)." 

What was the third diagnostic test? Or do you mean each diagnostic test had three possible result categories? I think the wording of "presumptive TB" is not the best for a CXR - I would categorise a CXR as "findings suggestive of TB". What was the means of bacteriological confirmation - culture, smear, molecular tests? Extrapulmonary TB can also be microbiologically confirmed - would these cases be classified as bacteriologically confirmed or others?

Line 153-155: "Data on LTFU and treatment success (TS) were assessed as outcomes where the results of treatment were classified as cure, completion, failure, death, or LTFU in the KNTSS database" Please rephrase for example "treatment success was categorized as cure, completion, failure, death, or LTFU as per the KNTSS database". 

Line 155-158: "The LTFU rate was defined as the sum of patients with unknown treatment results (marked in the database as 'not evaluated') and interrupted treatment (marked as 'interrupted') over a total number of new TB patients registered on KNTSS within each designated cohort year." Please consider editing, for examples "Patients were defined as LTFU if they had unknown treatment outcomes or interrupted treatment." Generally a rate has an time element to it (usually the denominator is person-years). What the sentence is describing is a proportion - i.e. the number of patients LTFU over the total number starting treatment

Line 159-161: "Those patients who were transferred to another institution (public or private) with more than months of treatment data missing post transfer were also marked as LTFU." - I am not sure I understand why this is mentioned separately. In a previous sentence it states that LTFU also include those with treatment interruption (it does not mention how long somebody who have to interrupt treatment to be classified as treatment interruption).

Line 169-172 "To assess the causal effect of the Korean PPM strategy on TB treatment outcomes, we used the difference-in-difference (DID) analysis with an assumption that, in absence of the policy intervention (or in this case, early policy intervention), unobserved differences between the compared groups would be the same over the 5-year period." My understanding of the study design was that the authors created one cohort of patients registered in 2009 and followed them up until treatment completion and one cohort registered in 2014 with follow-up until treatment completion. The study does not span a period - it provides two different snapshots. 

Line 186-188 "PSM was done to address potential biases that may be associated with the confounding factors arising from imbalances in the observed covariates from our non-longitudinal and non-counterfactual nature of our comparator groups." Please rephrase "PSM was performed to address potential biases that may be associated with the confounding factors arising from imbalances in the observed covariate because of the non-longitudinal and non-counterfactural nature of the analysis."

Results

Line 200-201 "patients managed in the public sector decreased (6,195 in 2009 → 2,803 in 2014)" please replace by (n=6195 (%) in 2009 and n=2803 (%) in 2014). Please consider doing this throughout the result section.

Line 212: please include p value

Line 214-216: is this truly rate and not proportion?

Line 219: "with both sectors" needs to be replace by "in both sectors"

Line 226: "higher age groups" please consider to replace by "older patients" or "in the older age-group"

Line 229-232 "Particularly, patients with transfer history were negatively impacted by the PPM program with 9.2% (95% CI, 6.3 to 12.1%) reduction in TSR and 6.7% (95% CI, 4.0 to 9.4%) increased LTFU rates in the private sector compared to the public sector patients in this subgroup over the 5-year period." Maybe consider splitting this sentence in two to make it clearer.

Lines 237- 241: "In crude DID analyses, certain patient sub-groups (patients with normal CXR results and EP-TB for TSR; and patients with EP-TB and unknown smear status for LTFU) had contradicting interpretation to that of PSM adjusted DID estimates which indicated that patients in these subgroups did not haves significant improvements in their treatment outcome as a result of the PPM." Please edit. 

Discussion

Line 253: replace "in" with "which"

Line 262-266: "Furthermore, alongside of PPM program expansion, continued expansion of NHI on TB service coverage and alignment of the comprehensive National Strategic Plan for TB control with relevant updates in government regulations and mandates have effectively lowered both financial and health systems barriers for patients suffering from TB-illness." Please consider rephrasing. 

Line 266: replace the word "eliminate"

Line 276: what is meant with policy blind spot?

Line 289-290: "This is because repeated measure of same population is not feasible from active TB cases." The authors could have used the population as a whole as a denominator. However, I think the problem is that one does not know what the denominator of the population is who access private/public care is. The authors may want to rephrase the sentence.

Figure 1: What does eligibility 100 TB notifications in a year mean? Explain either in the legend or in the figure.

Figure 3: I do not understand the purpose of this figure - a bit more explanation may be needed (i.e. a legend)?

Figure 5: The reasons why the TSRU needs to be explained. There is no x-axis. Rather than using abbreviations I would write the meaning of the categories at the left side of the figure. 

Reviewer #3: This paper seeks to study the effect of placing publicly funded nurses in private hospitals on TB-related outcomes in South Korea.

I was unable to follow the methods on the difference-in-differences study design and analysis. In particular, the unit of analysis was not clear. If the PPM intervention is the "treatment", then it seems the authors have taken the private (versus public) sector as the treatment unit - that is, in the data the unit that receives the intervention is the private sector. With two time periods, the analysis is effectively making use of four data points. This number of units is simply too small to do an analysis when done at the group level. The alternative, of performing an analysis on individual or patient level data, while adjusting standard errors for clustering in the standard ways, is not valid with such a small number of units. 

It is like doing a randomised controlled trial with two units in which one is assigned to the treatment group and the other to the control group. Any analysis of such a study would need to account for the small number of clusters. One approach is to analyse the data at the cluster level but there are other more sophisticated methods based on individual level data, see: https://www.ncbi.nlm.nih.gov/pubmed/29025158. The same basic problem applies to a DiD analysis in which there is a small number of units. The easiest approach is to analyse the data at the cluster-level (I'm not aware that the other methods described in the reference above can be used with a DiD setup). 

Please note that it is not so difficult to find published studies that analyse data at the individual level when the number of units is very small (including a PLOS Med study in which there are two units at the level of intervention assignment and the data are analysed at the individual level without any adjustment for clustering, let alone an adjustment for clustering using the standard, incorrect, methods). These studies are flawed and the data should have been analysed at the cluster (group) level. 

Without knowing the data better, it is hard for me to suggest a way forward. It does look, however, that the best one can do is to show descriptive results of the trends (as is done), without any statistical tests or regression analysis. Obviously such findings come with huge caveats and limitations. 

A second issue is that there is no great reason to believe the parallel trends assumption (as required for a valid DiD approach) would hold in this study. To help the reader, the authors should present trend data showing that the pre-trends (the trends before the PMM started) in the outcomes were similar between public and private sectors.

[LINK]

---

## [Decision Letter · Decision Letter 1]

24 Jan 2021

Dear Dr. Choi,

Thank you very much for submitting your revised manuscript "Impact Evaluation of Nationwide Public-Private Mix (PPM) Program for Tuberculosis under National Health Insurance: A Difference-In-Difference Analysis of the South Korean PPM" (PMEDICINE-D-19-03769R1) for consideration at PLOS Medicine. We do apologize for the long delay in sending you a decision. 

Your paper was evaluated by the editors and sent to independent reviewers, including a statistical reviewer and two new reviewers as some of the original reviewers were unavailable. The reviews are appended at the bottom of this email and any accompanying reviewer attachments can be seen via the link below:

[LINK]

In light of these reviews, we will not be able to accept the manuscript for publication in the journal in its current form, but we would like to invite you to submit a further revised version that fully addresses the reviewers' and editors' comments. You will appreciate that we cannot make a decision about publication until we have seen the revised manuscript and your response, and we expect to seek re-review by one or more of the reviewers. 

We hope to receive your revised manuscript by Feb 12 2021 11:59PM. Please email us (plosmedicine@plos.org) if you have any questions or concerns.

Please let me know if you have any questions. Otherwise, we look forward to receiving your revised manuscript in due course. 

Sincerely,

Richard Turner, PhD

rturner@plos.org

Please briefly explain the reasons that full access cannot be provided to study data, in your data statement. 

Please also add the URL for KCDC, or a point of contact via email. 

Please restructure your title by moving the country name before the colon (e.g., "... in South Korea: a difference ..."). 

In your abstract, please remove "highly" from "highly privatized". 

Please quote aggregate demographic details for study participants in your abstract.

Please trim your author summary. The three individual subsections should each consist of 3-4 points, each consisting of no more than 1-2 short sentences each. 

Please ensure that you refer to the attached STROBE checklist in the Methods section ("See S1_STROBE_Checklist" or similar). 

At line 381 and any other instances, please add "to our knowledge" if you wish to claim "the first", for example. 

Please quote exact p values or "p<0.001" throughout the paper, unless there are specific statistical reasons to quote smaller p values. 

Please substitute "sex" for "gender" throughout the text, where appropriate. 

We suggest "difference in differences", throughout. 

Please remove the information on study funding and competing interests from the end of the main text. In the event of publication, this information will appear in the article metadata via entries in the submission form. 

Please revisit your reference list to ensure that all citations meet journal style. For example, 6 author names should be listed rather than 3, followed where appropriate by "et al.". All italics and boldface text should be converted to plain text.

Please abbreviate journal names as appropriate, and, noting reference 30, ensure that all citations include full access information. 

Please adapt the attached STROBE checklist so that individual items are referred to by section (e.g., "Methods") and paragraph number rather than by page or line numbers, as the latter generally change in the event of publication. 

Comments from the reviewers:

*** Reviewer #1: 

The authors have addressed all my points except for one small matter.

They have now clarified the coefficents but I also suggested these could be presented together rather than scattered through the manuscript. I still think this is a good idea but I cannot seem to find it. Have I missed something?

Michael Dewey

*** Reviewer #4: 

Review of " Impact Evaluation of Nationwide Public-Private Mix (PPM) Program for Tuberculosis under National Health Insurance: A Difference-In-Difference

Analysis of the South Korean PPM"

This paper provides an interesting analysis on an important topic. I only have a few minor comments on the paper:

1. The authors refer to the WHO goals on Tuberculosis. They might consider mentioning Sustainable Development Goal #4 (which includes sub-goals for TB as well) as part of the motivation.

2. The authors mention that PPM-dedicated nurses provide patient education and treatment adherence. What is included in the patient education? While this may not be so important for the statistical analysis of treatment success (or is it?), it would be useful to understand what the program does in this respect.

3. The paper mentions that patients with drug-resistant TB were removed from the sample. How many are there of these in the data? Also, it would be useful to know how big of a problem drug resistant TB is in South Korea.

4. There is a literature in economics which analyses the impact of historical TB policies using differences-in-difference type techniques (e.g. Anderson et al., (American Economic Journal: Applied Economics, 2018); Egedesø et al. (Economic Journal, 2020), Clay et al. (Journal of Development Economics, 2020). Given the focus on TB and the overlap in technique, I think these papers deserve to be cited in the paper.

5. Table 1 gives information on how the treated and the control group differ in terms of observable characteristics. The table looks somewhat messy. I would suggest stating the total subsample for the treated and non-treated and then show everything else in percentage terms, e.g. the percentage of men.

6. S2 Figure A provides support for the parallel trend assumption. From 1993 to 2009, the trends look approximately parallel. Yet, from 1987 to 1993, something seems to be going on in the private sector. Is there any policy that could explain this? Parallel trends seem a reasonable assumption given the long period in the 1990s and 2000s in which the development looks parallel, so I do not think that is a big problem, but it would be useful to make some comment on this.

*** Reviewer #5: 

Review of Yu et al

M/s: Impact Evaluation of Nationwide Public-1 Private Mix (PPM) Program for Tuberculosis under National Health Insurance: A Difference-In-Difference Analysis of the South Korean PPM

Note: Although this is a revised manuscript, and includes responses to previous reviewers, this is the first time that the current reviewer has seen this manuscript.

Summary

This manuscript used a difference-in-difference (DID) analysis and compared the changes in treatment outcome (treatment success (TS) and loss to followup (LTFU)) between the public and private sector TB patient cohorts in 2009 and 2014. The baseline covariates examined were: age, gender, nationality, transfer history (treatment reported at one or more institutions since reported as a TB case), geographic division (categorized as metropolitan, city, and towns designated), and diagnostic test results (chest X-ray, smear, and culture). The primary result being claimed is that the PPM intervention applied only to the private sector patients resulted in a faster improvement trend/achievement for the private sector patients vs public sector patients.

Overall assessment

On the positive side, this is an important topic in public health (how to improve quality in private sector provision), the Korean approach is impressive and critical to document, and it is good to see a more rigorous than usual analysis of programmatic TB data. 

On the negative side, the result may not be greatly surprising, given the treatment success data available from KCDC at the start of the study, and there are some questions around how the results are analyzed and presented (see below for the 2 major comments - which the authors may well be able to respond to successfully).

Major comments

1) There is a lack of clarity in describing the main results.

Page 59-60, line 350: "Although TS improved in both sectors by 2014 (to 93.4% and 83.9% respectively in 2014), improvements in the private sector (Figure 4 A & B, red lines) was at least 4.1% (p value<0.0001 with 95% Confidence Interval [CI], 2.9 to 5.3%) higher than that of the public sector (Figure 4 A & B, blue lines)."

First off, this is referring to Fig 3, not Fig 4.

Second, it is really difficult for the reader to understand: (a) where the 4.1% in the text comes from (what is it referring to; how do you derive that from Fig 3); and (b) what we are actually meant to conclude from Fig 3. Most readers from the TB side (including this one) are not used to interpreting DID scores, or at least need them to be more clearly explained, including an explanation of which parts of Fig 3 we are meant to be comparing.

Same set of issues for the 8.7% in the next paragraph related to LTFU.

I would suggest you need something really simple such as: "The total percentage of increase in TS between 2009 and 2014 (red lines, Fig 3B) was greater for private sector patients (0.839-0.703=0.136) than for public sector patients (0.93-.836=0.094), leading to a positive DID of 4.2% (0.136-0.094=0.042, or 4.2%)." 

This is not interpretation of the results (which would belong in the discussion) but explanation of how to read Fig 3 (which belongs properly in the Results section, which is what I'm referring to here). The same kind of fix is needed for the following paragraph about LTFU.

2) My biggest question: why is DID the right and most fair way to make this comparison? I've leave the final determination about statistical methods to reviewers 1 and 3, who seem much more knowledgeable on this. But I have a point based on logic that I hope those reviewers can also assess. 

That point goes as follows: What I care about most as a TB person is the rate at which unfavorable treatment outcomes are being replaced by favorable treatment outcomes. As noted in the first paragraph of the discussion, the "gaps in TS and LTFU" are what we really care about. So, what is the rate of improvement in these gaps? 

Therefore, instead of DID, I can use Fig 3B to look at the proportion of unfavorable LTFU outcomes that are being eliminated between 2009 and 2014. By 2014, public sector manages to get rid of (0.145-0.053)/0.145=63% of their original, 2009 proportion of unfavorable LTFU outcomes. In the same period, private sector gets rid of (0.252-0.073)/0.252=71% of their original unfavorable outcomes. So private sector does indeed get rid of a higher proportion of the poor outcomes that they started with in the 2009 cohort. To me, that would be a clearer and more intuitive way to present these same results.

For TS, we can do a similar analysis. But here we ask what proportion of ALL unfavorable outcomes (1-TS, which includes LTFU and other poor outcomes such as death) were eliminated over time. This calculation for public sector is: ((1-.836)-(1-.93))/(1-.836)=(0.164-0.07)/0.164= 57% of all unfavorable outcomes from 2009 were eliminated by the time we got to 2014.

And the same calculation for the private sector: ((1-.703)-(1-.839))/(1-.703)= (.297-.161)/.297= 46%.

So actually in terms of getting rid of all unfavorable outcomes, the proportion of improvement was worse for the private sector than it was for the public sector (even though the absolute gain was greater for private, since its original deficit was so much bigger). If my calculations are correct, this somewhat undermines the central message of the paper, and at least needs to be acknowledged.

Minor comments

a) Page 51, line 152: "formation of global working groups at the Stop TB"

should be

"formation of a global PPM working group hosted by the Stop TB Partnership"

b) Page 51, line 159: "As such, there currently lacks long-term evidence of fully scaled-up national PPM program(s)"

Apart from fixing the grammar, this is a defensible statement, and indeed the authors may wish to cite https://pubmed.ncbi.nlm.nih.gov/30926734/ to support this claim. 

So I'm OK with the statement remaining. But in addition it would only be fair to add an acknowledgement of the very significant expansion of PPM in India. India does not have fully national coverage as of yet, and the outcomes are not yet fully clear - hence the authors' claim being defensible. But considering the far, far greater size of both the population and the TB burden in India relative to South Korea, this huge PPM effort in India should at least be acknowledged. I suggest citing the most recent Joint Monitoring Mission for the India national TB program in support of this; it can be found at https://tbcindia.gov.in/showfile.php?lid=3536 (it has an extensive section on PPM progress).

c) Page 59, line 340: "More non-Korean citizen, however, sought TB care in the public sector in both years"

I think what you actually mean is: "The proportion of the patients who were non-Korean citizens was higher in the public sector cohort than in the private sector cohort in both years…."

d) Page 60, line 366: "[people in the private sector and with transfer history…] did not demonstrate to have improvements in TS as a result of the PPM program between 2009 and 2014"

This is incorrect, since these are DID values, not absolute improvement values.

It should read: "For certain patient groups - [over 60, those with transfer history, etc etc] - any improvement in TS for private sector patients was no greater than any improvement in TS for public sector patients, so there was not a completely clear impact from the PPM program between 2009 and 2014."

The same issue comes up on page 62, line 418: "elderly (60 or older) and culture confirmed patients managed in the private sector clearly did not benefit from the PPM program". Not true. They just didn't improve as rapidly as the improvements in the public sector cohort.

e) Page 62, line 410: "as many high TB burden countries with emerging economies shift towards

411 universal health coverage (UHC),"

This somewhat mixes the concept of national health insurance (NHI, and/or social health insurance (SHI)), which was the topic of the paragraph up to this point, with UHC. Obviously, those two concepts are not the same (NHI is one potential way of trying to achieve UHC). Maybe make that clearer. And consider including the following citation on the topic of TB and SHI/NHI: Wells WA, Stallworthy G, Balsara Z. How tuberculosis programs can navigate the world of social health insurance. Int J Tuberc Lung Dis. 2019. 23:26-37. doi: 10.5588/ijtld.18.0289.

f) Page 65, line 485: "to integrate key private sectors" should be "to integrate key private sector providers"

***

[LINK]

---

## [Editor Report · Decision Letter 2]

21 Jun 2021

Dear Dr. Choi,

Thank you very much for re-submitting your manuscript "Evaluating the Impact of the Nationwide Public-Private Mix (PPM) Program for Tuberculosis under National Health Insurance in South Korea: A Difference in Differences Analysis" (PMEDICINE-D-19-03769R2) for consideration at PLOS Medicine. We do apologize for the long delay in sending you a response. 

I have discussed the paper with editorial colleagues and our academic editor and I am pleased to tell you that, once the remaining editorial and production issues are fully dealt with, we expect to be able to accept the paper for publication in the journal.

[LINK]

Please let me know if you have any questions, and we look forward to receiving the revised manuscript shortly.   

Sincerely,

Richard Turner, PhD

rturner@plos.org

Requests from Editors:

At line 53, please make that "... to date.".

Around line 65, please quote summary information on sex and average age for the two cohorts. 

We think that an additional short sentence is needed immediately prior to line 70, quoting the percentages for TS in private and public sectors at the two time points (as at line 355).

At line 70, please reword "As a result ..." to avoid making a claim of causality in this observational study, e.g., to "After ..."; and remove the word "vastly".

Please add "in South Korea" around line 80.

At line 119, please replace "effectively reduced", implying a causal interpretation, with "... was associated with a reduction" or similar, as appropriate for the observational research design. 

Please confirm that there were no data-driven changes to the analysis plan.

At line 393, for example, please avoid the claim of causality implicit in "... was central in improving ...", and instead state "was associated with an improvement in patient outcomes ..." or similar.

Please make similar adaptations to any other elements of text reflecting a causal interpretation. 

Please remove the word "greatly" at line 396. Again, "... as a result" implies causality and should be amended. 

To the sentence beginning at line 397 ("Improvements in treatment outcomes ...") please add the caveat that the outcomes studied remained superior in the public sector. 

At line 459, should that be "... may not have had an equal ..."?

Please use the style "... 5 years" throughout the text, although numbers should be written out at the start of sentences. 

Throughout the text, please remove spaces from within the reference call-outs, e.g., "... (Figure 1A) [18,19].".

In the figures and tables, and any other instances in the ms, please convert "p<0.0001" to "p<0.001".

In the first two rows of the STROBE checklist, right hand column, we suggest simply specifying "title" and "abstract" rather than giving the page numbers. 

***

---

## [Editor Report · Decision Letter 3]

28 Jun 2021

Dear Dr. Choi,

Thank you very much for re-submitting your manuscript "Evaluating the Impact of the Nationwide Public-Private Mix (PPM) Program for Tuberculosis under National Health Insurance in South Korea: A Difference in Differences Analysis" (PMEDICINE-D-19-03769R3) for consideration at PLOS Medicine.

Before we are able to proceed further, we will need to ask you to address the remaining issues listed at the end of this email.

In revising the manuscript for further consideration here, please ensure you address the specific points made by the editors. In your rebuttal letter you should indicate your response to the reviewers' and editors' comments and the changes you have made in the manuscript. Please submit a clean version of the paper as the main article file. A version with changes marked must also be uploaded as a marked up manuscript file.

Please let me know if you have any questions, and we look forward to receiving the revised manuscript shortly.   

Kind regards,

Richard Turner, PhD

rturner@plos.org

Requests from Editors:

In the data statement, you provide a web address for Statistics Korea, which has a "contact" section. Are you able to also provide a specific contact address (institutional email address preferred) at KCDA for readers interested in inquiring about access to study data?

At line 69 (abstract) the second "private" can be removed.

At line 75, please make that "loss to follow-up".

Please add a new final sentence to the "Methods and findings" subsection of your abstract, which should begin "Study limitations include ..." or similar and should quote 2-3 of the study's main limitations. 

At line 117, should that be "reduction in ..."?

At line 358, we suggest "indicates" rather than "infers".

***

---

## [Editor Report · Decision Letter 4]

30 Jun 2021

Dear Dr Choi, 

On behalf of my colleagues and the Academic Editor, Dr Murray, I am pleased to inform you that we have agreed to publish your manuscript "Evaluating the Impact of the Nationwide Public-Private Mix (PPM) Program for Tuberculosis under National Health Insurance in South Korea: A Difference in Differences Analysis" (PMEDICINE-D-19-03769R4) in PLOS Medicine.

PRESS

Sincerely, 

Richard Turner, PhD 

rturner@plos.org